# Increased transmissibility of SARS-CoV-2 lineage B.1.1.7 by age and viral load

Frederik Plesner Lyngse [1,2,3 ✉], Kåre Mølbak [3,4], Robert Leo Skov [3], Lasse Engbo Christiansen [5], Laust Hvas Mortensen[6,7], Mads Albertsen [8], Camilla Holten Møller[3], Tyra Grove Krause[3], Morten Rasmussen[3], Thomas Yssing Michaelsen [8], Marianne Voldstedlund[3], Jannik Fonager [3], Nina Steenhard[3], The Danish Covid-19 Genome Consortium* & Carsten Thure Kirkeby[4]

New lineages of SARS-CoV-2 are of potential concern due to higher transmissibility, risk of severe outcomes, and/or escape from neutralizing antibodies. Lineage B.1.1.7 (the Alpha variant) became dominant in early 2021, but the association between transmissibility and risk factors, such as age of primary case and viral load remains poorly understood. Here, we used comprehensive administrative data from Denmark, comprising the full population (January 11 to February 7, 2021), to estimate household transmissibility. This study included 5,241 households with primary cases; 808 were infected with lineage B.1.1.7 and 4,433 with other lineages. Here, we report an attack rate of 38% in households with a primary case infected with B.1.1.7 and 27% in households with other lineages. Primary cases infected with B.1.1.7 had an increased transmissibility of 1.5–1.7 times that of primary cases infected with other lineages. The increased transmissibility of B.1.1.7 was multiplicative across age and viral load.

[1] Department of Economics & Center for Economic Behaviour and Inequality, University of Copenhagen, Copenhagen, Denmark. [2] Danish Ministry of Health, Copenhagen, Denmark. [3] Statens Serum Institut, Copenhagen, Denmark. [4] Department of Veterinary and Animal Sciences, Faculty of Health and Medical Sciences, University of Copenhagen, Copenhagen, Denmark. [5] DTU Compute, Lyngby, Denmark. [6] Statistics Denmark, Copenhagen, Denmark. [7] Department of Public Health, University of Copenhagen, Copenhagen, Denmark. [8] Department of Chemistry and Bioscience, Aalborg University, Aalborg, Denmark. *A list of authors and their affiliations appears at the end of the paper. ✉email: fpl@econ.ku.dk

Control of the current pandemic caused by Severe Acute Respiratory Syndrome Coronavirus 2 (SARS-CoV-2) is increasingly challenged by the emerging variants of concern (VOC). These include lineages associated with increased transmissibility[1–3], severe outcomes such as hospitalization[4,5], and/or mortality[6,7] and/or whether they can escape immune protection by natural immunization[8]. Variants, such as the Alpha variant of SARS-CoV-2 VOC 202012/01 (also known as clade 20I/501Y.V1 or lineage B.1.1.7), are identified by whole genome sequencing (WGS)[3]. The B.1.1.7 lineage was first identified in the southeast of England in September 2020[3]. Since then, it spread quickly to other countries, and became a dominant strain in large parts of the world[9,10]. In Denmark, B.1.1.7 was first detected on November 14, 2020, and by March 2021 comprised more than 90% of the circulating lineages, but was in the summer of 2021 replaced by the even more transmissible Delta variant (lineage B.1.617.2)[11]. As a consequence of the increased transmissibility of lineage B.1.1.7, nonpharmaceutical interventions (NPIs), such as physical distancing and other restrictions, have been shown to be less effective for sustaining epidemic control, and vaccination uptake to reach herd immunity is projected to be higher[12].

Increased transmissibility of B.1.1.7 was estimated in models that use data from community based surveillance with limited metadata. The estimated increased transmissibility of B.1.1.7 range from 35% to 130% across countries[13–16]. In Denmark, it was estimated to be 36–55% higher than other circulating lineages[17,18]. These estimates are sensitive to country-specific conditions, such as other circulating lineages, implemented NPIs, and contact tracing efforts, which can all affect the generation time.

Most studies of B.1.1.7 transmission have not addressed transmission in specific settings, e.g., households, and have not included detailed explanatory variables known to affect transmissibility, such as age of primary cases, age of exposed individuals, and viral load of primary case.

Household members live close together and typically share kitchen, bathroom, and common rooms. Thus, close contact is difficult to limit within households, and may present a challenge for epidemic control. Therefore, studies of transmission in the household domain serve as an opportunity to learn about transmission patterns. Furthermore, household transmission may serve as a bridge between otherwise separate transmission domains, such as schools and physical workplaces, despite implemented NPIs.

Denmark has one of the highest SARS-CoV-2 real-time reverse transcription polymerase chain reaction (RT-PCR) testing and WGS capacities in the world. Furthermore, tests for SARS-CoV-2 are free of charge and testing is widespread with current levels of testing exceeding 30,000 weekly tests per 100,000 persons. Moreover, there is comprehensive social insurance, and SARS-CoV-2 sick leave is fully reimbursed. Hence, neither access to tests nor financial reasons represent major barriers to obtaining a test. Since December 2020, it has been a government policy to use WGS data for surveillance of the Danish epidemic. This has resulted in more than 70% of all RT-PCR positive tests being selected for WGS since January 11, 2021.

The aim of this study was to estimate the household transmissibility SARS-CoV-2 for lineage B.1.1.7 compared with other lineages, by age and viral load. Furthermore, we wanted to estimate whether there is a multiplicative or additive effect of the increased transmissibility of B.1.1.7 compared with other lineages.

## Results

Within the study period, a total of 8,093 household primary cases were identified, of which 82% (6,632) were selected for WGS, and 65% (5,241) generated a high-quality SARS-CoV-2 genome (Table 1). Lineage B.1.1.7 was found in 15% (808) of these genomes. The primary cases lived in households comprising 2-6 persons with a total of 16,612 potential secondary cases, of which

**Table 1 Summary Statistics.**

| | Primary Cases | | | | Potential Secondary Cases | Positive Secondary Cases | Attack Rate (%) | (95%-CI) |
|---|---|---|---|---|---|---|---|---|
| | Total | Selected for WGS | With Genome | With B.1.1.7 | | | | |
| Total | 8093 | 6632 | 5241 | 808 | 16,612 | 4133 | 25 | (24–26) |
| **Sex** | | | | | | | | |
| Male | 3648 | 3013 | 2406 | 419 | 8905 | 2190 | 25 | (24–26) |
| Female | 4445 | 3619 | 2835 | 389 | 7707 | 1943 | 25 | (24–26) |
| **Age** | | | | | | | | |
| 0–10 | 419 | 327 | 237 | 54 | 3490 | 822 | 24 | (22–25) |
| 10–20 | 795 | 670 | 557 | 91 | 3270 | 755 | 23 | (21–25) |
| 20–30 | 1531 | 1294 | 1020 | 204 | 2347 | 494 | 21 | (19–23) |
| 30–40 | 1353 | 1101 | 870 | 143 | 1876 | 483 | 26 | (24–28) |
| 40–50 | 1464 | 1182 | 920 | 119 | 2167 | 521 | 24 | (22–26) |
| 50–60 | 1443 | 1166 | 917 | 132 | 2020 | 536 | 27 | (25–29) |
| 60–70 | 669 | 539 | 449 | 38 | 919 | 315 | 34 | (31–37) |
| 70–80 | 300 | 255 | 190 | 23 | 392 | 150 | 38 | (33–43) |
| >80 | 119 | 98 | 81 | <5 | 131 | 57 | 44 | (37–54) |
| **Household Size** | | | | | | | | |
| 2 | 3308 | 2716 | 2108 | 298 | 3308 | 1019 | 31 | (29–32) |
| 3 | 1886 | 1549 | 1235 | 189 | 3635 | 843 | 23 | (22–25) |
| 4 | 1848 | 1486 | 1178 | 193 | 5368 | 1292 | 24 | (23–26) |
| 5 | 790 | 661 | 534 | 92 | 3042 | 714 | 23 | (21–26) |
| 6 | 261 | 220 | 186 | 36 | 1259 | 265 | 21 | (17–25) |

Notes: This table provides summary statistics for the number of primary cases, potential secondary cases, positive secondary cases, and attack rates in the study, stratified by sex, age and household sizes Summary statistics for five-year age groups are shown in Table S2. Summary statistics stratified by the primary cases are shown in Table S3 and S4. 95%-Confidence intervals are clustered on the household level.

**Table 2 Intra-household correlation of lineages between primary and positive secondary cases.**

| Primary cases | | Positive secondary cases | | | | Potential secondary | Attack Rate | |
|---|---|---|---|---|---|---|---|---|
| Lineage | N | B.1.1.7 | Other lineages | No Genome | Total | cases | (%) | (95%-CI) |
| B.1.1.7 | 808 | 472 | 19 | 165 | 656 | 1719 | 38 | (35–41) |
| Other lineages | 4433 | 18 | 1750 | 721 | 2489 | 9115 | 27 | (26–28) |
| No Genome | 2852 | 133 | 540 | 315 | 988 | 5778 | 17 | (16–18) |
| Total | 8093 | 623 | 2309 | 1201 | 4133 | 16,612 | 25 | (24–26) |

Notes: There were 8093 primary cases, of which 808 (10%) where infected with B.1.1.7, 4433 (55%) were infected with other lineages, and 2852 (35%) did not have a successfully sequenced genome. The 808 primary cases infected with B.1.1.7 had 656 positive secondary cases. Of these cases, 75% (472 + 19 = 491) were successfully sequenced. Of these, 96% (472) were infected with B.1.1.7 and 4% (19) with other lineages. 95%-Confidence intervals are clustered on the household level.

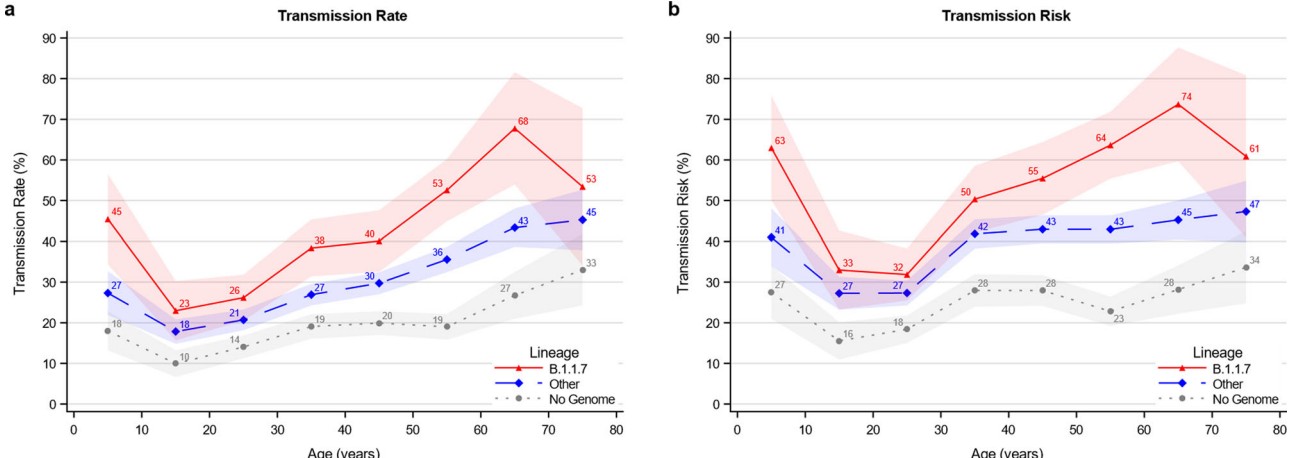

**Fig. 1 Age structured transmissibility stratified by lineage of the primary case. a** The transmission rate describes the proportion of potential secondary cases within the household that were infected. **b** The transmission risk describes the proportion of infected primary cases that infected at least one secondary case. Figure S7 provides the same graphs for five-year age groups. The markers show the estimates of the mean. The shaded areas show the 95% confidence bands of the estimates clustered on the household level.

4,133 tested positive. This implies an attack rate of 25% (4,133/16,612).

The intra-household correlation of lineages between primary and positive secondary cases was investigated using the proportion of positive secondary cases that were infected with the same lineage (B.1.1.7 vs. other lineages) as the primary case (Table 2). For primary cases infected with B.1.1.7, 96% of the positive secondary cases (that were successfully sequenced) were also infected with B.1.1.7. Similarly, for primary cases infected with other lineages, 99% of the positive secondary cases (that were successfully sequenced) were also infected with other lineages. For the primary cases without a successfully sequenced genome, 20% of the positive secondary cases (that were successfully sequenced) were infected with B.1.1.7 and 80% with other lineages. This distribution roughly corresponds to the underlying prevalence in the community during period of the study.

In households where the primary cases were infected with B.1.1.7, the attack rate was 38%, compared with 27% when the primary cases were infected with other lineages, and 17% when the primary case did not have a successfully sequenced genome.

The age specific transmissibility roughly followed a J-shaped pattern with the lowest transmission from primary cases in the 10 to 30 years age range, higher from younger children, and highest from elderly cases (Fig. 1). Both the transmission rate (Fig. 1, a) and the transmission risk (Fig. 1, b) were higher for B.1.1.7 (red) compared with other lineages (blue) across all ten-year age groups. The transmissibility was lower for primary cases without a successfully sequenced genome (gray).

To investigate whether the increased transmissibility of B.1.1.7 compared with other lineages was best described as an additive or multiplicative effect, we compared the model fit of both a linear and a logistic regression analysis. We compared the fit of the two models using the Akaike Information Criteria (AIC) and found that the logit model was a better fit (Supplementary Note 3). This supports the hypothesis that the effect of the increased transmissibility is best described as a multiplicative effect.

Using a logit model, we estimated the increased transmission rate and transmission risk for B.1.1.7 compared with other lineages. In Table 3, we present the crude estimates as well as models controlling for age of the primary case, age of the potential secondary cases, and Ct value of the primary case. Primary cases infected with B.1.1.7 were 1.5 times more transmissible than primary cases infected with other lineages, without any adjustments. When controlling for age and viral load, this effect was 1.6.

## Discussion

We used national population data to estimate the household transmissibility of the SARS-CoV-2 lineage B.1.1.7 compared with other lineages. We utilized detailed administrative register data comprising the full Danish population and the ability to link data across registers on a person level. This combined with a large proportion of the population being tested, a large national WGS capacity, and an understanding of the sampling selection process, allowed us to estimate the household transmissibility controlling for age and viral load.

**Table 3 Odds ratio estimates for transmissibility for B.1.1.7 compared with other lineages.**

| | Transmission Rate | | | | Transmission Risk | | |
|---|---|---|---|---|---|---|---|
| | I | II | III | IV | V | VI | VII |
| B.1.1.7 | 1.50 | 1.68 | 1.69 | 1.63 | 1.52 | 1.66 | 1.61 |
| 95%-CI | (1.30–1.72) | (1.46–1.94) | (1.47–1.95) | (1.39–1.91) | (1.31–1.77) | (1.42–1.93) | (1.36–1.90) |
| Constant | ✓ | ✓ | ✓ | ✓ | ✓ | ✓ | ✓ |
| Age, Primary Case | | ✓ | ✓ | ✓ | | ✓ | ✓ |
| Age, Pot. Sec. Case | | | ✓ | ✓ | | | ✓ |
| Ct Value | | | | ✓ | | | ✓ |
| Observations | 10,834 | 10,834 | 10,834 | 8762 | 10,834 | 10,834 | 8762 |
| Households | 5241 | 5241 | 5241 | 4172 | 5241 | 5241 | 4172 |

We found that B.1.1.7 had a household transmissibility 1.5–1.7 times higher compared with other lineages circulating at the time of the study, which is in line with B.1.1.7 transmissibility estimates from modelling studies[3,4,7,19].

Furthermore, we estimated the transmissibility across age groups and found that lineage B.1.1.7 generally follows the pattern of other lineages, where teenagers are the least transmissible within households. However, B.1.1.7 was consistently more transmissible per age group compared with other lineages.

We found that the increased transmissibility of B.1.1.7 was a multiplicative effect of the transmissibility of other lineages, rather than an additive effect. Only one previous study has estimated both the additive effect and the multiplicative effect[20], but they did not test the two models against each other. The multiplicative effect implies that the known risk factors for increased transmissibility are amplified by 1.5-1.7 times when the case is infected with B.1.1.7.

We have previously found that younger children are more transmissible within the household compared with teenagers[21,22]. There is still disagreement about the effect of B.1.1.7 on the transmissibility in children[23,24]. In this study, we found that children (<10 years)—like adults—also exhibit a higher transmissibility within households if they are infected with B.1.1.7. However, children were generally still less transmissible than persons aged 60 and above, who were most transmissible (Fig. 1). This pattern is affected by the behavior of people, and could reflect that couples often sleep together, increasing the risk of transmission. Furthermore, the age profile of the transmissibility may have implications for the decision of vaccinating children in the future. Indeed, the transmissibility of children under five years of age to other household members is not negligible, and this aspect may have even further ramifications where the Delta variant has been the dominant strain.

The increased transmissibility of 1.5–1.7 times for B.1.1.7 may have additional public health implications. For example, for contact tracing, this means that cases with a high predicted transmissibility, e.g., by viral load or age[22,25,26], that are infected with highly transmissible variants are even more transmissible and thus should be prioritized. Naturally, household contacts are different from other contacts. They are more frequent, closer and of a longer time duration, compared with exposures in other settings, such as workplaces. Additionally, many people live with a partner around their own age and parents live with their children. The results underline the importance of timely and efficient management and isolation of confirmed cases to limit transmission in the household domain. Transmission in households may serve as a bridge between otherwise separate domains, such as schools and physical workplaces, despite implemented NPIs in these domains. Moreover, it might be more challenging for young children to maintain social distancing and to adhere to NPIs in general, more outbreaks of highly transmissible strains in

kindergardens and primary schools could be expected. Furthermore, our results imply that the transmissibility of B.1.1.7—and possibly other successful lineages—should be modelled as a multiplicative effect and not an additive effect. This is pivotal for the validity and accuracy of simulation models of the current pandemic, which are used as tools for decision makers, but need further studies.

The mechanisms behind the increased transmissibility of B.1.1.7 are not fully elucidated. It has been suggested that enhanced binding of the N501Y mutated spike protein may result in increased binding affinity to the human angiotensin-converting enzyme 2 (ACE2)[27,28]. Furthermore, Kissler et al.[29] and Calistri et al.[30]. found that the infectious period for cases infected with B.1.1.7 was generally longer compared with cases infected with other lineages. For children, this has also previously been described for seasonal influenza by Ng et al.[31]. The longer infectious period could contribute to the increased transmissibility of lineage B.1.1.7.

There are several strengths in the present study. This nation-wide study was based on detailed administrative data that enabled us to control for individual specific characteristics of both primary and potential secondary cases. We restricted our sample to only include households with 2-6 members during a period with no national holidays, no changes in government restrictions, and systematic sampling for WGS. Furthermore, we challenged our approach by investigating the intra-household correlation of lineages between primary and positive secondary cases. We found that the vast majority of secondary cases were infected with the same lineage (B.1.1.7 vs other lineages) as the primary case. When investigating the intra-household correlation of lineages between primary and positive secondary cases, we found that 96% of the secondary cases associated with a primary case infected with B.1.1.7 were also infected with B.1.1.7. Similarly, we found that 1% of the secondary cases associated with a primary case infected with other lineages were infected with B.1.1.7. This suggests that only a minor fraction of the positive secondary cases were misclassified. As there were several lineages circulating in the study period, this increases confidence that the data reflects household transmission rather than community transmission.

We estimated the increased transmissibility of B.1.1.7 relative to a baseline of other circulating lineages at the time of the study. It is evident that these estimates depend on the composition of this baseline. In our study period, 82% of all positive cases were selected for WGS and 65% of the total cases had a successfully sequenced genome (Tabel S1). Furthermore, we restricted our study period to an interval where B.1.1.7 was present, but not fully dominating. This enabled us to obtain transmissibility estimates from both B.1.1.7 and other strains at the same time. The commonly circulating lineages included B.1.258.11, B.1.258, B.1.221.3, B.1.221. B.1.160 and B.1.177 (Fig. S1), i.e., before B.1.617.2 (Delta) became dominant. Therefore, it is not likely that

our findings are an artefact generated by a misleading baseline of other lineages. It would be difficult to repeat the study at the present, comparing B.1.1.7 with B.1.617.2 (Delta) due to the fact that vaccination uptake was rising when the Delta variant became dominant.

There was a significant proportion of positive RT-PCR positive samples without a successfully sequenced genome that could not be assigned to specific lineages. This can potentially result in sample selection bias. Samples with low viral load (high Ct values) were less likely to be selected for WGS and successfully sequenced (Fig. S3 and S4). Cases with low viral load have been shown to be less transmissible (Figs. S10 and S11)[22,25,26]. If cases infected with B.1.1.7 have higher viral loads than cases infected with other lineages, this would lead to over-sampling of cases infected with B.1.1.7. Presently, this is not fully elucidated. Calistri et al.[30] have found that cases infected with B.1.1.7 have a higher viral load, whereas Kissler et al.[29] and the present study (Fig. S7) found no difference. This implies that over-sampling of cases infected with B.1.1.7 was not a problem in this study. Furthermore, we controlled for Ct values in our multivariable regression model, and this confirmed that B.1.1.7 was associated with increased transmission even after adjusting for Ct values.

There were only relatively minor changes in the estimates of the increased transmissibility of B.1.1.7 compared with other lineages when varying the controls (Table 3). This suggests that the increased transmissibility of B.1.1.7 is independent of the age of the infected person, age of the exposed person and Ct value. Moreover, the estimates could be sensitive to the definition of primary and secondary cases. However, when we restricted our analysis to only include secondary cases identified on days 1–14, 2-14, 3-14, and 4-14, we found no significant changes in the estimates. The same was true, when we excluded households with co-primary cases. This demonstrates that the estimates of the increased transmissibility of B.1.1.7 were not dependent on the inclusion criteria for secondary cases nor definition of co-primary cases.

Some limitations apply to this study. This is a retrospective observational study, therefore causality naturally cannot be inferred. Additionally, we did not have access to data on rapid antigen tests, which have been increasingly used in Denmark since December 2020. All cases with a positive antigen test were recommended to have a confirmatory RT-PCR test. If cases tested positive with an antigen test and not a RT-PCR test, we could not include these as positive cases. Moreover, uncertainties regarding the heterogeneous transmissibility across lineage B.1.1.7 and other circulating lineages are present, as we did not have data on symptoms and exposure history. Thus, the estimates here represent the general picture across all households.

Despite of these limitations, we believe that the results of this study provide useful new insights into the transmissibility of B.1.1.7.

In summary, we found an attack rate of 38% in households with a primary cases infected with B.1.1.7 and 27% in households with a primary case infected with other lineages. Primary cases infected with B.1.1.7 had an increased transmissibility of 1.5-1.7 times that of primary cases infected with other lineages. The increased transmissibility of B.1.1.7 is multiplicative across age and viral load.

The spread of lineage B.1.1.7 has been explosive in countries across the world. The results found in this study add new knowledge that can be used to understand transmission patterns of highly successful strains in the household domain, which serves as an important and often neglected arena of transmission. Further studies are needed to evaluate the transmissibility in other settings, such as workplaces, schools and other places of infection.

## Methods

**Register data**. We used comprehensive Danish register data, comprising the full population of Denmark, all RT-PCR tests for SARS-CoV-2 from the Danish Microbiology Database (MiBa), and all positive RT-PCR tests that were sampled for WGS. The RT-PCR test results included the cycle threshold (Ct) value, which reflects the viral load of the sample. Thus, a low Ct value implies that the sample contained a high viral load. We used the Danish civil registration number, which is a unique personal identifier, to link positive and negative RT-PCR tests to a national registry of address codes. Thereby, we established a data set of all Danish households, which enabled analysis of presumed household transmission by age, Ct value and SARS-CoV-2 lineage; as also conducted in Lyngse et al.[21]

In Supplementary Note 1, we provide descriptive statistics from December 20, 2020 (week 52) to February 21, 2021 (week 7) to provide background information for our choice of study period.

**Study data**. We restricted our study sample to comprise primary cases identified in the period from January 11 (week 2) to February 7, 2021 (week 5). We allowed for 14 days follow up for secondary cases to occur. There were no changes in public health measures or COVID-19 related restrictions in this period, and the period did not include any public holidays. Week 52 (2020) and week 1 (2021) were affected by Christmas and New Year, while schools opened for grades 0–4 (ages 6–10 years) in week 6. We further restricted our study sample to households with two to six members in order to have relatively comparable households, and thus we excluded, e.g., long-term care facilities and other residential institutions. During the study period, the population were tested due to a large variety of reasons, including being able to attend work with a negative test. Contact tracing is one of the main non-pharmaceutical interventions in Denmark and thus primary cases in this study could be discovered due to centralized or independent contact tracing efforts, symptoms, and/or screening. Persons identified with infection were asked to self-isolate within the household, however, in many cases self-isolation was not fully possible, e.g., due to lack of space, number of bathrooms, or caregiving needs of children.

**Whole genome sequencing (WGS)**. During the study period, RT-PCR tests for SARS-CoV-2 could be obtained from either community testing facilities at Test-Center Denmark (TCDK) or in hospitals, which serve patients and healthcare personnel. All samples from TCDK were analyzed at Statens Serum Institut (SSI), whereas samples from hospitals were analyzed at the hospitals' departments of clinical microbiology. Testing through TCDK accounted for approximately 75% of all tests and 70% of all positive tests in Denmark[22]. Furthermore, TCDK has used the same protocol for RT-PCR across the full study period. Sequencing of the genome of SARS-CoV-2 was carried out by The Danish COVID-19 Genome Consortium, which was established in March 2020 with the purpose of assisting public health authorities by providing rapid genomic monitoring of the spread of SARS-CoV-2. Positive tests were sequenced from both hospitals and TCDK. No laboratory analyses, including WGS, were performed for this study.

As not all positive samples have been selected for WGS, it is important to understand the sample selection process. Information on WGS sample selection criteria and Ct values was only available for positive cases that were identified through TCDK. On January 11, 2021 (week 2), SSI started systematic selection of positive samples for WGS using a Ct value cut-off, in order to maximize the probability of a suitable genome for WGS analysis. During week 2, SSI used a cut-off of Ct < 30, Ct < 32, and Ct < 35. In week 3-6, SSI used a cut-off of Ct < 35. During periods with excess WGS capacity, SSI included samples with higher Ct values (35 < Ct < 38). An RT-PCR test was considered positive, if Ct ≤ 38. This is supported by the data (Figs. S3 and S4).

**Sample selection bias**. In our data, not all positive cases have a successfully sequenced genome. This can be due to various reasons, e.g., sequencing capacity constraints. Moreover, the probability of successfully sequencing a genome is correlated with the viral load, which is reflected in the Ct value. Therefore, sample selection bias is a major concern. If some cases have a higher probability of being selected for WGS than others, it can lead to false conclusions. In Supplementary Note 1, we provide summary statistics to substantiate our choice of study period. As both viral load (Ct values) and age of the primary case are associated with transmissibility[22,25,26], we explored this.

**Statistical analyses**. We defined primary cases as the first identified RT-PCR positive SARS-CoV-2 case in a household, and any cases that were detected in the same household within the following 1–14 days were considered to be secondary cases (see also sensitivity analysis of this below). If more than one person tested positive on the first date, the primary case was randomly selected. We utilized two concepts for transmissibility of the primary case: transmission risk and transmission rate. The transmission risk describes the risk of infecting at least one other person within the household, and equals one if any (one or more) secondary cases are identified within the same household, and zero otherwise. The transmission rate is the proportion of potential secondary cases within the same household that tested positive. The two transmissibility measures are weighted on the primary case level, such that each primary has a weight of one.

Furthermore, we utilized one concept for susceptibility of the potential secondary case: attack rate. The (secondary) attack rate is defined as the proportion of potential secondary cases that tested positive. The attack rate is weighted on the potential secondary case level, such that each potential secondary case has a weight of one.

We estimated the transmission rate and transmission risk for each 10 year age group separately and stratified by lineage B.1.1.7 and other lineages, using a generalized linear regression model.

To investigate whether the increased transmissibility of B.1.1.7 compared with other lineages was best described as an additive or multiplicative effect, we compared the model fit of both a linear and a logistic regression analysis, using the Akaike Information Criteria (AIC).

We used a logistic regression model to estimate the odds ratio of the transmission rate and transmission risk for B.1.1.7 compared with other lineages. As the transmissibility can be dependent on the age of the primary case, the age of the potential secondary case, and the viral load (measured by cycle threshold (Ct) value)[22,25,26], we included these as explanatory variables.

See Supplementary Note 3 for further details of the statistical analyses.

We used SAS 9.4 to manage and analyze the data.

*Sensitivity analyses.* To investigate the robustness of the estimated transmissibility across age groups, we supplemented our main analyses of ten-year age groups with five-year age groups.

We estimated the transmission rate and transmission risk by Ct value intervals.

The estimates are sensitive to the definition of primary and secondary cases. In our approach, it is possible that a co-primary case may be misclassified as a secondary case, if she is tested positive one or more days later than the first identified case. In order to investigate the robustness of the results to the definition of primary and secondary cases, we additionally analyzed the data defining secondary cases as those that tested positive at 1-14 days (as in the main analysis), 2-14 days, 3-14 days and 4-14 days after the primary case. We furthermore performed our main analyses excluding households with co-primary cases (405 households, 5%) to investigate the sensitivity to misclassification of primary cases.

**Ethical statement**. This study was conducted on administrative register data. According to Danish law, ethics approval is not needed for such research. All data management and analyses were carried out on the Danish Health Data Authority's restricted research servers with project number FSEID-00004942. The publication only contains aggregated results and no personal data.

The publication is, therefore, not covered by the European General Data Protection Regulation.

**Reporting summary**. Further information on research design is available in the Nature Research Reporting Summary linked to this article.

## Data availability
The data used in this study are available under restricted access due to Danish data protection legislation. The data are available for research upon reasonable request to The Danish Health Data Authority and Statens Serum Institut and within the framework of the Danish data protection legislation and any required permission from Authorities. We performed no data collection and performed no sequencing for this study. The Danish public health authorities deposits all SARS-CoV-2 sequences to GISAID (https://www.gisaid.org/), including those used in this study. The GISAID accession identifiers are listed in Supplementary File S1.

## Code availability
The code used for this study is deposited at GitHub: https://github.com/Flyngse/SARS-CoV-2_B.1.1.7_Transmissibility.git.

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

## Acknowledgements

We thank Statens Serum Institut and The Danish Health Data Authority for data access and helpful institutional knowledge. We also thank the rest of the Expert Group for Mathematical Modelling of COVID-19 at Statens Serum Institut. Lastly, we acknowledge the authors, originating and submitting laboratories of the sequences from GISAID's EpiCov Database. All submitters of data may be contacted directly via the GISAID website www.gisaid.org. Funding: Frederik Plesner Lyngse: Independent Research Fund Denmark (Grant no. 9061-00035B.); Novo Nordisk Foundation (grant no. NNF17OC0026542); the Danish National Research Foundation through its grant (DNRF-134) to the Center for Economic Behavior and Inequality (CEBI) at the University of Copenhagen. Laust Hvas Mortensen: Novo Nordisk Foundation (NNF17OC0027594, NNF17OC0027812). Mads Albertsen: Poul Due Jensen Fonden (Corona Danica); Styrelsen for Forskning og Uddannelse, Forskning i COVID-19 (0238-00002B Forståelse af smitteveje gennem sekventering og analyse af coronavirus genomer).

## Author contributions

F.P.L. performed all data analysis. F.P.L., C.T.K., and K.M. wrote the first draft. F.P.L., C.T.K., K.M., R.B.L., L.E.C., L.H.M., M.A., C.H.M., T.G.K., M.R., T.Y.M., M.V., J.F., and N.S. contributed to the discussion and writing the final draft.

## Competing interests

The authors declare no competing interests.

## Additional information

## The Danish Covid-19 Genome Consortium

**AAU Coordination** Mads Albertsen[8]

**AAU Laboratory** Jakob Brandt[8], Simon Knuttson[8], Emil A. Sørensen[8], Thomas B. N. Jensen[8], Trine Sørensen[8], Celine Petersen[8], Clarisse Chiche-Lapierre[8], Frederik T. Hansen[8], Emilio F. Collados[8], Amalie Berg[8], Susanne R. Bielidt[8], Sebastian M. Dall[8], Erika Dvarionaite[8], Susan H. Hansen[8], Vibeke R. Jørgensen[8], Trine B. Nicolajsen[8], Wagma Saei[8] & Stine K. Østergaard[8]

**AAU Bioinformatics** Thomas Y. Michaelsen[8], Vang Le-Quy[8], Mantas Sereika[8], Rasmus H. Kirkegaard[8], Kasper S. Andersen[8] & Martin H. Andersen[8]

**AAU CLAUDIA/IT** Karsten K. Hansen[8], Mads Boye[8], Mads P. Bach[8], Peter Dissing[8], Anton Drastrup-Fjordbak[8], Michael Collin[8] & Finn Büttner[8]

**AAU Legal and Admin** Susanne Andersen[8] & Lea Sass Otte[8]

**AAU SUND** Martin Bøgsted[8] & Rasmus Brøndum[8]

**AAU Computer Science** Katja Hose[8], Tomer Sagi[8] & Miroslav Pakanec[8]

**Statens Serum Institut** Anders Fomsgaard[3], Morten Rasmussen[3], Søren M. Karst[3], Jannik Fonager[3], Vithiagaran Gunlan[3], Marc Bennedbæk[3], Raphael Sieber[3], Kirsten Ellegaard[3], Anna C. Ingham[3], Thor B. Johannesen[3], Martin Basterrechea[3], Berit Lilje[3], Kim L. Ng[3], Sofie M. Edslev[3], Sharmin Baig[3],

Marc Stegger[3], Povilas Matusevicius[3], Lars Bustamante Christoffersen[3], Man-Hung Eric Tang[3], Christina Wiid Svarrer[3], Nour Saad Al-Tamimi[3], Marie Bækvad-Hansen[3], Jonas Byberg-Grauholm[3], Mette Theilgaard Christiansen[3], Karen Mare Jørgensen[3], Nicolai Balle Larsen[3] & Arieh Cohen[3]

**Aalborg University Hospital** Henrik Krarup[8], David Fuglsang-Damgaard[8], Mette Mølvadgaard[8] & Marc T. K. Nielsen[8]

**Rigshospitalet** Kristian Schønning[9], Martin S. Pedersen[9], Rasmus L. Marvig[9] & Nikolai Kirkby[9]

**Hvidovre Hospital** Uffe V. Schneider[10], Jose A. S. Castruita[10], Nana G. Jacobsen[10] & Christian Ø. Andersen[10]

**Aarhus University Hospital** Mette Christiansen[11], Ole H. Larsen[11], Kristian A. Skipper[11], Søren Vang[11], Kurt J. Handberg[11], Carl M. Kobel[11], Camilla Andersen[11], Irene H. Tarpgaard[11] & Svend Ellermann-Eriksen[11]

**Odense University Hospital** Marianne Skov[12] & Thomas V. Sydenham[12]

**Herlev Hospital** Lene Nielsen[13], Line L. Nilsson[13], Martin B. Friis[13], Thomas Sundelin[13] & Thomas A. Hansen[13]

**Sygehus Lillebælt** Anders Jensen[14] & Ea S. Marmolin[14]

**Zealand University Hospital** Xiaohui C. Nielsen[15] & Christian H. Schouw[15]

**Sydvestjysk Sygehus** John E. Coia[16] & Dorte T. Andersen[16]

[9]Rigshospitalet, Copenhagen, Denmark. [10]Hvidovre Hospital, Hvidovre, Denmark. [11]Aarhus University Hospital, Aarhus, Denmark. [12]Odense University Hospital, Odense, Denmark. [13]Herlev Hospital, Herlev, Denmark. [14]Sygehus Lillebælt, Kolding, Denmark. [15]Zealand University Hospital, Køge, Denmark. [16]Sydvestjysk Sygehus, Esbjerg, Denmark.

