## [Peer Review File · Nature Communications]

Increased Transmissibility of SARS-CoV-2 Lineage B.1.1.7 by Age and Viral LoadREVIEWER COMMENTS

Reviewer #2 (Remarks to the Author):

This contribution presents analyses of the transmissibility of SARS-CoV-2 by infected individuals in Danish households, and of differentials in transmissibility by Coronavirus lineage. It demonstrates the degree of excess transmissibility of SARS-Cov-2 lineage B.1.1.7 relative to other lineages. The main strength of the study is that it relies on data from a context with broad coverage of testing as well as whole genome sequencing, and a situation where tested individuals can be linked to their (non-infected) household members via Denmark's comprehensive population-register system. The study provides novel information on (i) the level of transmissibility of the virus, which amounts to an "attack rate" of about 27-38% for people who share a household, and (ii) the differential transmissibility between different lineages of the virus as observed in Denmark in early 2021.

Key concepts are well explained and abbreviations are spelled out, with the exception of the Ct concept that is presented without further notice.

The study brings analyses of the risks of primary cases to infect at least one other household member, as well as that of other household members being infected by a primary case in his or her household. The latter concept is defined as an "attack rate", where the proportion of infected household members is analyzed weighted on the potentially secondary case level. The concepts of "risks" and "rates" may sometimes be used a bit different in different scientific setups and I have no objections to the chosen definition of an "attack rate". The authors could perhaps still consider validating it with the analysis of an individual "risk" of getting infected by a primary case?

Naturally, not all positive cases are followed up with a genome sequencing and there are a few steps of selection that may cause some bias in the cases that end up being included in the analytical design. The first step is that of the PCR test being analyzed at the TestCenter Denmark, a condition that could be spelled out more explicitly at p5. As far as I can judge, the authors provide a good rationale for how the selection of cases is unlikely to produce any serious biases in the analyses that are presented. The cases that had not led to any genome sequencing had a much lower transmissibility, at 17%, than those cases that went through genome sequencing, which is commented on in the final discussion.

In terms of substance findings, the authors would better refer to the age profile of transmissibility as being J- rather than U-shaped. And to pay better attention to the strongly elevated transmissibility of the virus among elderly household members than to the less elevated risk of spreading the virus for small children. The authors could also discuss the fact that the majority of household members were after all not infected by a positive case in the household, which could perhaps be regarded as slightly surprising.

A few temporal statements in the text could be added as the spread of and different lineages of the virus is still something of a moving target.

There are some missing words in the sentence that stretches over rows 150-153.

Reviewer #3 (Remarks to the Author):

Households are important venues for SARS-CoV-2 transmission when widespread community control measures are in place, which urges prioritization of studies on transmissibility and risk factors for household SARS-CoV-2 transmission. As SARS-CoV-2 variant B.1.1.7 continues to spread across the world, there is still a lack of knowledge regarding data-based household secondary attack rate and associated risk factors for the transmission of SARS-CoV-2 variant B.1.1.7. In this context, the

authors used comprehensive administrative data from Denmark, comprising the full population, all SARS-CoV-2 RT-PCR tests, and all whole genome sequencing lineage data, to estimate household transmissibility by age, lineage and viral load. This would provide valuable information for understanding how the mutations affect the transmissibility of SARS-CoV-2. But I would argue that the methods and results of this manuscript need to be improved.

Major comments:

1. How could the authors rule out community transmission within households?
2. Besides the date of RT-PCR positive for each infector, is there any other rules that were used to define primary and secondary cases? If not, I suggest authors define the generation of transmission for each infector by considering the combination of exposure history and key transmission parameters (e.g., incubation period), and then estimate the transmissibility by generation of transmission, lineages, age and Ct values.
3. I suggest authors add more details regarding the implementation of non-pharmaceutical interventions and COVID-19 surveillance system during study period in Denmark. In particular:
 - 1) Was the primary case identified through symptom-based surveillance? Were clusters with pre-symptomatic and asymptomatic SARS-CoV-2 transmission involved in this study?
 - 2) Was the primary case "removed" from the household after the confirmation?
 - 3) Did all potential secondary cases receive RT-PCR testing regularly?
4. Is there any differences in epidemiological characteristics (e.g., age distribution) between cases who were selected for WGS or not? If yes, do the authors believe their conclusions are robust enough? Please also state whether there is a difference in epidemiological characteristics (e.g., age and clinical severity) between samples obtained from hospitals and TCDK?
5. Besides epidemic lineage, age and viral load of an infector, there are several other factors that might be associated with SARS-CoV-2 transmission, e.g., levels of exposure to an infector, generation of transmission and clustering effect attributed to an infector and a household. I am concerned that the model specifications in Table S4 and Table S5 may be not comprehensive, and this may affect the accuracy of estimated transmissibility of the variant B.1.1.7. In addition, authors failed to report details regarding methods and results of generalized linear regression model, such as the definition and regression coefficients of each covariate, model fit statistics, and hypothesis testing results.
6. In Figure 1, the transmission rate and transmission risk among different age groups might be confounded by other factors, e.g., age of their contacts. Please add age-specific transmission matrix, including: 1) number of primary cases and their potential secondary cases by age group, and 2) number of primary cases and their secondary cases by age group.
7. The visualization in this paper needs to be improved.
8. Please specify the proportion of households with "co-primary cases" and please repeat this random process for several times to account for the uncertainty of transmission.

Minor comments:

1. I suggest the authors add 95% confidence interval for secondary attack rates in Table 1, Table 2 and Table S2, and give coefficients with the p-value and odds ratio (OR) for each covariate in Table 3.
2. The title of figures should be placed at the bottom of each figure
3. "SARSCoV-2" on page 5 line 109 should revised as "SARS-CoV-2".
4. "Supplementary Table S5" on page 8 line 178 should be revised as "Supplementary File".

Response to Reviewer comments

Reviewer #2 (R2)

Reviewer #2 (Remarks to the Author):

R2: This contribution presents analyses of the transmissibility of SARS-CoV-2 by infected individuals in Danish households, and of differentials in transmissibility by Coronavirus lineage. It demonstrates the degree of excess transmissibility of SARS-Cov-2 lineage B.1.1.7 relative to other lineages. The main strength of the study is that it relies on data from a context with broad coverage of testing as well as whole genome sequencing, and a situation where tested individuals can be linked to their (non-infected) household members via Denmark's comprehensive population-register system. The study provides novel information on (i) the level of transmissibility of the virus, which amounts to an "attack rate" of about 27-38% for people who share a household, and (ii) the differential transmissibility between different lineages of the virus as observed in Denmark in early 2021.

R2: Key concepts are well explained and abbreviations are spelled out, with the exception of the Ct concept that is presented without further notice.

AU: We have added a short explanation of Ct values in line 87-89: "The RT-PCR test results included the cycle threshold (Ct) value, which reflects the viral load of the sample. Thus, a low Ct value implies that the sample contained a high viral load."

R2: The study brings analyses of the risks of primary cases to infect at least one other household member, as well as that of other household members being infected by a primary case in his or her household. The latter concept is defined as an “attack rate”, where the proportion of infected household members is analyzed weighted on the potentially secondary case level. The concepts of “risks” and “rates” may sometimes be used a bit different in different scientific setups and I have no objections to the chosen definition of an “attack rate”. The authors could perhaps still consider validating it with the analysis of an individual “risk” of getting infected by a primary case?

AU: We agree that the definitions are not always clear and therefore we took care in defining concepts have defined it clearly in the manuscript. The attack rate is a well known concept in epidemiology. It mirrors the incidence of illness in a defined population who is exposed to a defined health event in a short time period. It is usually expressed as a proportion. In the present context it is a measure of illness in the susceptible exposed population is an expression of the susceptibility to infection, given that you live in a household with an infected person. Thus, we define the “risk” for potential secondary cases getting infected by a primary case is in this manuscript defined as the “attack rate”: “The (secondary) attack rate is defined as the proportion of potential secondary cases that tested positive.” (line 153-156). Thus, we believe this is covered in the analyses. We have chosen to focus on transmissibility and not susceptibility. However, we do agree that susceptibility heterogeneity is important to further investigate.

The transmission rates and transmission risk are not standard concepts in epidemiology, but were for the present study defined as measures of the probability that a primary case were successful in spreading the virus. Again, this also relates to the susceptibility of the exposed population, but also many other factors including behavior, contact patterns and biology. It will be beyond the scope to discuss these issues in detail, but it is important to be clear about the definitions.

R2: Naturally, not all positive cases are followed up with a genome sequencing and there are a few steps of selection that may cause some bias in the cases that end up being included in the analytical design. The first step is that of the PCR test being analyzed at the TestCenter Denmark, a condition that could be spelled out more explicitly at p5. As far as I can judge, the authors provide a good rationale for how the selection of cases is unlikely to produce any serious biases in the analyses that are presented. The cases that had not led to any genome sequencing had a much lower transmissibility, at 17%, than those cases that went through genome sequencing, which is commented on in the final discussion.

AU: We are sorry that our explanation has been vague. We include all tests performed in Denmark—both from hospitals and TestCenter Denmark. However, we only have Ct values on the tests performed at TestCenter Denmark. We have added a clarification in line 122: “Positive tests were sequenced from both hospitals and TCDK.”

R2: In terms of substance findings, the authors would better refer to the age profile of transmissibility as being J- rather than U-shaped. And to pay better attention to the strongly elevated transmissibility of the virus among elderly household members than to the less elevated risk of spreading the virus for small children. The authors could also discuss the fact that the majority of household members were after all not infected by a positive case in the household, which could perhaps be regarded as slightly surprising.

AU: We agree that the pattern is better described as a J shape, and have changed the U-shape to a J-shape. Line 235.

We have also added some reflections of the cause of this pattern, and how it might change in the future. Line 290-296: “However, children were generally still less transmissible than persons aged 60 and above, who were most transmissible (Figure 1). This pattern is affected by the behavior of people, and could reflect that couples often sleep together, increasing the risk of transmission. Furthermore, the age profile of the transmissibility may have implications for the decision of vaccinating children in the future. Indeed, the transmissibility of children under five years of age to other household members is not negligible, and this aspect may have even further ramifications where the delta variant has been the dominant strain.”

For the sake of brevity, we have not further discussed the result that the majority of potential secondary cases were not infected.

R2: A few temporal statements in the text could be added as the spread of and different lineages of the virus is still something of a moving target.

AU: We agree that the different lineages are moving targets. Since the manuscript was submitted, the epidemiological situation has changed in most countries. B.1.1.7 (the alpha variant) is no longer dominant but by and large has been replaced with B.1.617.2 (the delta variant). Furthermore, household transmission will additionally change due to vaccination uptake; this will reduce susceptibility among potential secondary cases. Nonetheless, we believe that our findings is of public health relevance, because (1) the work reflects a conceptual model for understanding variant-related transmission differences; this should be generalizable to other strains as well, (2) adds further to the understanding of the epidemiology of B.1.1.7 in particular and SARS-CoV-2 in general, and (3) underscores the role of the household domain in SARS-CoV-2 transmission, including the non-negligible transmission from small children to other household members.

We have updated parts of the introduction and the discussion to include these aspects.

R2: There are some missing words in the sentence that stretches over rows 150-153.

AU: We have clarified this. Line 164-167: “As the transmissibility can be dependent on the age of the primary case, the age of the potential secondary case, and the viral load (measured by cycle threshold (Ct) value)^{20,21,22}, we included these as explanatory variables.”

Reviewer #3 (R3)

Reviewer #3 (Remarks to the Author):

R3: Households are important venues for SARS-CoV-2 transmission when widespread community control measures are in place, which urges prioritization of studies on transmissibility and risk factors for household SARS-CoV-2 transmission. As SARS-CoV-2 variant B.1.1.7 continues to spread across the world, there is still a lack of knowledge regarding data-based household secondary attack rate and associated risk factors for the transmission of SARS-CoV-2 variant B.1.1.7. In this context, the authors used comprehensive administrative data from Denmark, comprising the full population, all SARS-CoV-2 RT-PCR tests, and all whole genome sequencing lineage data, to estimate household transmissibility by age, lineage and viral load. This would provide valuable information for understanding how the mutations affect the transmissibility of SARS-CoV-2. But I would argue that the methods and results of this manuscript need to be improved.

AU: Thank you for the overall positive assessment. We are very delighted and agree with the reviewer that these results are important and can contribute to the policy response to the ongoing pandemic. We hope that our answer below will help convince you of the robustness of our results.

Major comments:

R3: 1. How could the authors rule out community transmission within households?

AU: In short, we cannot. However, we believe that the probability of acquiring infection from the community is generally much lower than the probability of being infected from an infector at home. Moreover, we chose a study period where many national restrictions were still at place, e.g., school closures, and after Christmas, where it is likely that many households interacted. Additionally, once a case is identified she and her household are expected per policy to isolate themselves from the community. Therefore, we believe the risk is negligible.

We try to address this concern in Table 2, where we found that 96% of secondary cases had the same lineage as the primary case, when the primary case was infected with B.1.1.7. This was estimated at a time, where many other lineages were present in the population and B.1.1.7 amounted to less than 25% of all cases (Figure S1). This indicates that infections most likely stem from household transmission.

Moreover, in Table S5, column IV, we only include secondary cases identified 4-14 days after the primary case. These cases are likely to have been isolated from the community and hence been infected in the household.

Lastly, we have further elaborated on this in the discussion. Line 343-345: “As there were several lineages circulating in the study period, this increases confidence that the data reflects household transmission rather than community transmission.”

R3: 2. Besides the date of RT-PCR positive for each infector, is there any other rules that were used to define primary and secondary cases? If not, I suggest authors define the generation of transmission for each infector by considering the combination of exposure history and key transmission parameters (e.g., incubation period), and then estimate the transmissibility by generation of transmission, lineages, age and Ct values.

AU: No other rules than the date of the RT-PCR positive test were used to define primary and secondary cases.

We thank the reviewer for the suggestion. However, this additional analysis would require new data that we do not have access to and a new set of assumptions that we cannot validate using the present data. For instance, we do not have access to the exposure history of cases, which strongly limits us in the definition of the generation of transmission.

R3: 3. I suggest authors add more details regarding the implementation of non-pharmaceutical interventions and COVID-19 surveillance system during study period in Denmark. In particular:

- 1) Was the primary case identified through symptom-based surveillance? Were clusters with pre-symptomatic and asymptomatic SARS-CoV-2 transmission involved in this study?
- 2) Was the primary case “removed” from the household after the confirmation?
- 3) Did all potential secondary cases receive RT-PCR testing regularly?

AU: We have now elaborated on the description of the situation in Denmark during the study period. Line 104-110: “During the study period, the population were tested due to a large variety of reasons, including being able to attend work with a negative test. Contact tracing is one of the main non-pharmaceutical interventions in Denmark and thus primary cases in this study could be discovered due to centralized or independent contact tracing efforts, symptoms, and/or screening. Persons identified with infection were asked to self-isolate within the household, however, in many cases self-isolation was not fully possible, e.g., due to lack of space, number of bathrooms, or caregiving needs of children.”

Furthermore, the possibility of isolation facilities outside the households, i.e., “corona hotels”, was available in all Danish municipalities. The stay was free-of-charge and included food. Less than 1% of cases in January 2021 used this offer.

R3: 4. Is there any differences in epidemiological characteristics (e.g., age distribution) between cases who were selected for WGS or not? If yes, do the authors believe their conclusions are robust enough? Please also state whether there is a difference in epidemiological characteristics (e.g., age and clinical severity) between samples obtained from hospitals and TCDK?

AU: We have added two graphs on the probability of the sample being selected for WGS by age (Figure S5), and by week (Figure S6). There is no clear evidence of a correlation between age and the probability of being selected for WGS during our study period (Figure S5 and table below). However, we do find a clear correlation with age prior to our study period. (To make Figure S3 comparable with Figure S5, we have changed Figure S3 to only include the study period.)

Persons tested in hospitals are a mix of patients and healthcare personnel (line 113-114). We do not have data on clinical severity or reasons for being tested. Lastly, all regressions including Ct values are excluding primary cases identified at hospitals, as we only have Ct values on tests obtained at TCDK. Hence, if there were large differences in the populations across testing sites (primary cases identified at hospitals vs. primary cases identified at TCDK), we would see large changes in the estimates, when excluding these (Table 3). We do not see this.

If we test the probability of being selected for WGS by age and testing site (TCDK vs. hospital), i.e., $\text{Prob}(\text{WGS selection}) = \text{AgeGroup} + \text{TCDK} * \text{AgeGroup} + \text{epsilon}$, where AgeGroup are 5-year age groups and TCDK is an indicator for being identified at TCDK, we find no significant interaction with TCDK. In particular, we find a p-value of 0.24 in the joint F-test. (However, if we included periods before our study period, we find a significant difference in the probability of being selected for WGS across age and testing place.) See table below. This supports that the selection to WGS is not age dependent in our study period.

Table: Test for significance of interaction between 5-year age groups and TCDK.

Study period		
	P-value	Degrees of Freedom
TCDK*Age_5	0.2362	20
Observations	18,592	
Overall period		
	P-value	Degrees of Freedom
TCDK*Age_5	<0.0001	20
Observations	66,530	

R3: 5. Besides epidemic lineage, age and viral load of an infector, there are several other factors that might be associated with SARS-CoV-2 transmission, e.g., levels of exposure to an infector, generation of transmission and clustering effect attributed to an infector and a household. I am concerned that the model specifications in Table S4 and Table S5 may be not comprehensive, and this may affect the accuracy of estimated transmissibility of the variant B.1.1.7. In addition, authors failed to report details regarding methods and results of generalized linear regression model, such as the definition and regression coefficients of each covariate, model fit statistics, and hypothesis testing results.

AU: We agree with the reviewer that there are several important factors related to SARS-CoV-2 transmissions that are not included in the reported analysis. Some of the factors mentioned are very difficult to obtain information on and were not available to us. We think that this is sufficiently clear from the description of the analyses given on page 7-8 and 53-55. The principles used in the regression modelling, including model search, is described in general terms in the methods section. We have further elaborated in the methods section, line 157-159: “We estimated the transmission rate and transmission risk for each 10 year age group separately and stratified by lineage B.1.1.7 and other lineages, using a generalized linear regression model.”. We have omitted parameter estimates and hypothesis testing results from the manuscript for the sake of brevity. We have included a list of the parameter estimates and test indices below. We are happy to include this in the appendix, but we feel that it would take up too much space in the main text of the manuscript.

Table S7: Linear Model excluding Age, Pot. Sec. Case and Ct values

Analysis Of Maximum Likelihood Parameter Estimates									
Parameter		DF	Estimate	Standard Error	Wald 95% Confidence Limits		Wald Chi-Square	Pr > ChiSq	
Intercept		1	0.1717	0.0164	0.1396	0.2038	109.90	<.0001	
B117_index		1	0.1066	0.0184	0.0706	0.1426	33.66	<.0001	
Age, Primary Case	Age_index_10	0-10	1	0.1165	0.0343	0.0493	0.1837	11.55	0.0007
	Age_index_10	20-30	1	0.0272	0.0207	-0.0133	0.0678	1.74	0.1873
	Age_index_10	30-40	1	0.0986	0.0223	0.0549	0.1424	19.54	<.0001
	Age_index_10	40-50	1	0.1253	0.0223	0.0817	0.1690	31.73	<.0001
	Age_index_10	50-60	1	0.1926	0.0228	0.1478	0.2373	71.10	<.0001
	Age_index_10	60-70	1	0.2735	0.0286	0.2174	0.3296	91.23	<.0001
	Age_index_10	70-80	1	0.2779	0.0395	0.2004	0.3554	49.39	<.0001
	Age_index_10	80-90	1	0.3914	0.0574	0.2788	0.5040	46.43	<.0001
	Age_index_10	10-20	0	0.0000	0.0000	0.0000	0.0000	.	.
	Scale		0	1.0000	0.0000	1.0000	1.0000		

Log Likelihood	-3,130
AIC	6,280

Observations	10,834
Clusters	5,241

Table S7: Logit Model excluding Age, Pot. Sec. Case and Ct values

Analysis Of Maximum Likelihood Parameter Estimates								
Parameter		DF	Estimate	Standard Error	Wald 95% Confidence Limits		Wald Chi-Square	Pr > ChiSq
Intercept		1	-1.5679	0.1104	-1.7843	-1.3515	201.64	<.0001
B117_index		1	0.5208	0.0818	0.3605	0.6810	40.58	<.0001
Age_index_10	0-10	1	0.6615	0.1781	0.3125	1.0106	13.80	0.0002
Age_index_10	20-30	1	0.1747	0.1331	-0.0861	0.4355	1.72	0.1893
Age_index_10	30-40	1	0.5699	0.1325	0.3102	0.8296	18.50	<.0001
Age_index_10	40-50	1	0.6988	0.1305	0.4430	0.9547	28.66	<.0001
Age_index_10	50-60	1	1.0009	0.1288	0.7485	1.2534	60.40	<.0001
Age_index_10	60-70	1	1.3399	0.1450	1.0556	1.6241	85.34	<.0001
Age_index_10	70-80	1	1.3542	0.1824	0.9967	1.7118	55.11	<.0001
Age_index_10	80-90	1	1.8168	0.2501	1.3266	2.3069	52.78	<.0001
Age_index_10	10-20	0	0.0000	0.0000	0.0000	0.0000	.	.
Scale		0	1.0000	0.0000	1.0000	1.0000		

Age, Primary Case

Log Likelihood	-3,128
AIC	6,277

Observations	10,834
Clusters	5,241

Table S7: Linear Model excluding Ct values

		DF	Estimate	Standard Error	Wald 95% Confidence Limits		Wald Chi-Square	Pr > ChiSq	
Age, Primary Case	Intercept	1	0.1878	0.0233	0.1421	0.2335	64.88	<.0001	
	B117_index	1	0.0958	0.0189	0.0588	0.1328	25.78	<.0001	
	Age_index_10	0-10	1	0.1118	0.0377	0.0378	0.1858	8.77	0.0031
	Age_index_10	20-30	1	0.0288	0.0260	-0.0221	0.0796	1.23	0.2681
	Age_index_10	30-40	1	0.0968	0.0264	0.0450	0.1485	13.44	0.0002
	Age_index_10	40-50	1	0.1186	0.0246	0.0704	0.1669	23.19	<.0001
	Age_index_10	50-60	1	0.1528	0.0260	0.1018	0.2038	34.48	<.0001
	Age_index_10	60-70	1	0.1956	0.0349	0.1272	0.2639	31.46	<.0001
	Age_index_10	70-80	1	0.1610	0.0541	0.0551	0.2670	8.87	0.0029
	Age_index_10	80-90	1	0.2318	0.0792	0.0765	0.3871	8.56	0.0034
	Age_index_10	10-20	0	0.0000	0.0000	0.0000	0.0000	.	.
Age, Pot. Sec. Case	Age_10	0-10	1	0.0069	0.0245	-0.0411	0.0549	0.08	0.7773
	Age_10	20-30	1	0.0156	0.0253	-0.0340	0.0653	0.38	0.5377
	Age_10	30-40	1	0.0204	0.0280	-0.0345	0.0753	0.53	0.4664
	Age_10	40-50	1	0.0278	0.0253	-0.0217	0.0774	1.21	0.2710
	Age_10	50-60	1	0.0519	0.0253	0.0023	0.1015	4.21	0.0402
	Age_10	60-70	1	0.0857	0.0333	0.0205	0.1510	6.63	0.0100
	Age_10	70-80	1	0.1193	0.0512	0.0191	0.2196	5.44	0.0197
	Age_10	80-90	1	0.1633	0.0795	0.0076	0.3191	4.22	0.0399
	Age_10	90-100	1	-0.3124	0.7960	-1.8726	1.2479	0.15	0.6948
	Age_10	10-20	0	0.0000	0.0000	0.0000	0.0000	.	.
	Scale		0	1.0000	0.0000	1.0000	1.0000		

Log Likelihood	-3,132
AIC	6,302

Observations	10,834
Clusters	5,241

Table S7: Logit Model excluding Ct values

	Parameter	DF	Estimate	Standard Error	Wald 95% Confidence Limits		Wald Chi-Square	Pr > ChiSq	
Age, Primary Case	Intercept	1	-1.7207	0.1334	-1.9821	-1.4592	166.41	<.0001	
	B117_index	1	0.5245	0.0819	0.3639	0.6850	40.98	<.0001	
	Age_index_10	0-10	1	0.7174	0.1837	0.3574	1.0774	15.25	<.0001
	Age_index_10	20-30	1	0.2133	0.1418	-0.0645	0.4912	2.26	0.1324
	Age_index_10	30-40	1	0.6327	0.1405	0.3573	0.9080	20.28	<.0001
	Age_index_10	40-50	1	0.7514	0.1325	0.4918	1.0111	32.17	<.0001
	Age_index_10	50-60	1	0.9218	0.1332	0.6608	1.1828	47.92	<.0001
	Age_index_10	60-70	1	1.1195	0.1606	0.8046	1.4343	48.57	<.0001
	Age_index_10	70-80	1	0.9523	0.2290	0.5034	1.4011	17.29	<.0001
	Age_index_10	80-90	1	1.2885	0.3281	0.6454	1.9316	15.42	<.0001
Age, Pot. Sec. Case	Age_index_10	10-20	0	0.0000	0.0000	0.0000	0.0000	.	.
	Age_10	0-10	1	0.0387	0.1194	-0.1954	0.2728	0.10	0.7459
	Age_10	20-30	1	0.0794	0.1226	-0.1610	0.3197	0.42	0.5174
	Age_10	30-40	1	0.1148	0.1345	-0.1489	0.3785	0.73	0.3934
	Age_10	40-50	1	0.1713	0.1216	-0.0671	0.4096	1.98	0.1591
	Age_10	50-60	1	0.2993	0.1160	0.0719	0.5266	6.66	0.0099
	Age_10	60-70	1	0.4620	0.1413	0.1851	0.7388	10.70	0.0011
	Age_10	70-80	1	0.6153	0.2074	0.2088	1.0218	8.80	0.0030
	Age_10	80-90	1	0.8178	0.3221	0.1864	1.4491	6.44	0.0111
	Age_10	90-100	1	-18.8063	13784.88	-27036.7	26999.06	0.00	0.9989
	Age_10	10-20	0	0.0000	0.0000	0.0000	0.0000	.	.
	Scale	0	1.0000	0.0000	1.0000	1.0000			

Log Likelihood	-3,118
AIC	6,273

Observations	10,834
Clusters	5,241

Table S7: Linear Model including Ct values

	Parameter		DF	Estimate	Standard Error	Wald 95% Confidence Limits		Wald Chi-Square	Pr > ChiSq
Age, Primary Case	Intercept		1	0.1435	0.0271	0.0903	0.1967	27.98	<.0001
	B117_index		1	0.0947	0.0201	0.0553	0.1342	22.14	<.0001
	Age_index_10	0-10	1	0.1322	0.0408	0.0522	0.2122	10.50	0.0012
	Age_index_10	20-30	1	0.0413	0.0255	-0.0087	0.0912	2.62	0.1055
	Age_index_10	30-40	1	0.1156	0.0262	0.0642	0.1670	19.44	<.0001
	Age_index_10	40-50	1	0.1480	0.0250	0.0990	0.1969	35.05	<.0001
	Age_index_10	50-60	1	0.1765	0.0263	0.1249	0.2281	44.90	<.0001
	Age_index_10	60-70	1	0.2264	0.0375	0.1529	0.2999	36.40	<.0001
	Age_index_10	70-80	1	0.2198	0.0691	0.0845	0.3552	10.13	0.0015
	Age_index_10	80-90	1	0.4071	0.1476	0.1178	0.6963	7.61	0.0058
Age, Pot. Sec. Case	Age_index_10	10-20	0	0.0000	0.0000	0.0000	0.0000	.	.
	Age_10	0-10	1	0.0092	0.0255	-0.0408	0.0592	0.13	0.7180
	Age_10	20-30	1	0.0189	0.0264	-0.0329	0.0706	0.51	0.4747
	Age_10	30-40	1	0.0159	0.0294	-0.0417	0.0735	0.29	0.5893
	Age_10	40-50	1	0.0405	0.0263	-0.0110	0.0920	2.38	0.1229
	Age_10	50-60	1	0.0667	0.0266	0.0146	0.1189	6.30	0.0121
	Age_10	60-70	1	0.0793	0.0367	0.0074	0.1512	4.68	0.0306
	Age_10	70-80	1	0.1120	0.0611	-0.0078	0.2318	3.36	0.0669
	Age_10	80-90	1	0.1575	0.1301	-0.0975	0.4125	1.47	0.2260
	Ct Value	Age_10	90-100	1	-0.3314	1.7325	-3.7271	3.0643	0.04
Age_10		10-20	0	0.0000	0.0000	0.0000	0.0000	.	.
CT_value_2		<18	1	0.2893	0.1550	-0.0145	0.5932	3.48	0.0620
CT_value_2		18-20	1	0.1571	0.0513	0.0566	0.2576	9.39	0.0022
CT_value_2		20-22	1	0.0519	0.0331	-0.0129	0.1167	2.46	0.1165
CT_value_2		22-24	1	0.0473	0.0264	-0.0044	0.0991	3.21	0.0732
CT_value_2		24-26	1	0.0123	0.0244	-0.0354	0.0601	0.26	0.6124
CT_value_2		26-28	1	-0.0109	0.0234	-0.0568	0.0350	0.22	0.6410
CT_value_2		30-32	1	-0.0239	0.0249	-0.0726	0.0249	0.92	0.3377
CT_value_2		32-34	1	-0.0136	0.0279	-0.0683	0.0412	0.24	0.6276
Ct Value	CT_value_2	34-36	1	-0.0137	0.0425	-0.0971	0.0696	0.10	0.7465
	CT_value_2	36-38	1	-0.1386	0.0648	-0.2657	0.0115	4.57	0.0326
	CT_value_2	28-30	0	0.0000	0.0000	0.0000	0.0000	.	.
	Scale		0	1.0000	0.0000	1.0000	1.0000		

Log Likelihood	-
AIC	2,452

Observations	8,762
Clusters	4,172

Table S7: Logit Model including Ct values

		DF	Estimate	Standard Error	Wald 95% Confidence Limits		Wald Chi-Square	Pr > ChiSq	
	Parameter								
	Intercept	1	-1.8454	0.1618	-2.1625	-1.5283	130.10	<.0001	
	B117_index	1	0.4875	0.0908	0.3096	0.6655	28.84	<.0001	
Age, Primary Case	Age_index_10	0-10	1	0.8243	0.2085	0.4157	1.2330	15.63	<.0001
	Age_index_10	20-30	1	0.2923	0.1528	-0.0072	0.5919	3.66	0.0558
	Age_index_10	30-40	1	0.7186	0.1516	0.4216	1.0157	22.48	<.0001
	Age_index_10	40-50	1	0.8727	0.1429	0.5926	1.1529	37.28	<.0001
	Age_index_10	50-60	1	1.0018	0.1424	0.7226	1.2810	49.46	<.0001
	Age_index_10	60-70	1	1.2050	0.1770	0.8580	1.5519	46.33	<.0001
	Age_index_10	70-80	1	1.1661	0.2876	0.6023	1.7298	16.43	<.0001
	Age_index_10	80-90	1	1.9802	0.6851	0.6373	3.3230	8.35	0.0038
	Age_index_10	10-20	0	0.0000	0.0000	0.0000	0.0000	.	.
	Age, Pot. Sec. Case	Age_10	0-10	1	0.0434	0.1328	-0.2168	0.3036	0.11
Age_10		20-30	1	0.0736	0.1368	-0.1945	0.3417	0.29	0.5903
Age_10		30-40	1	0.0827	0.1507	-0.2126	0.3780	0.30	0.5830
Age_10		40-50	1	0.2383	0.1334	-0.0232	0.4998	3.19	0.0740
Age_10		50-60	1	0.3616	0.1274	0.1119	0.6113	8.06	0.0045
Age_10		60-70	1	0.4224	0.1602	0.1083	0.7364	6.95	0.0084
Age_10		70-80	1	0.5609	0.2470	0.0769	1.0450	5.16	0.0231
Age_10		80-90	1	0.7180	0.5220	-0.3051	1.7411	1.89	0.1690
Age_10		90-100	1	-17.5826	16847.67	-33038.4	33003.24	0.00	0.9992
Age_10		10-20	0	0.0000	0.0000	0.0000	0.0000	.	.
Ct Value	CT_value_2	<18	1	1.2420	0.6488	-0.0296	2.5137	3.66	0.0556
	CT_value_2	18-20	1	0.7507	0.2213	0.3169	1.1846	11.50	0.0007
	CT_value_2	20-22	1	0.2828	0.1541	-0.0191	0.5848	3.37	0.0664
	CT_value_2	22-24	1	0.2584	0.1253	0.0128	0.5040	4.25	0.0392
	CT_value_2	24-26	1	0.0691	0.1212	-0.1683	0.3066	0.33	0.5682
	CT_value_2	26-28	1	-0.0353	0.1176	-0.2657	0.1951	0.09	0.7641
	CT_value_2	30-32	1	-0.1371	0.1307	-0.3933	0.1190	1.10	0.2941
	CT_value_2	32-34	1	-0.0746	0.1460	-0.3607	0.2115	0.26	0.6093
	CT_value_2	34-36	1	-0.0795	0.2234	-0.5174	0.3583	0.13	0.7219
	CT_value_2	36-38	1	-0.9033	0.4327	-1.7515	-0.0552	4.36	0.0368
	CT_value_2	28-30	0	0.0000	0.0000	0.0000	0.0000	.	.
	Scale		0	1.0000	0.0000	1.0000	1.0000		

Log Likelihood	-2,448
AIC	4,953

Observations	8,762
Clusters	4,172

Table S8: Linear Model excluding Ct values

Analysis Of Maximum Likelihood Parameter Estimates									
Parameter		DF	Estimate	Standard Error	Wald 95% Confidence Limits		Wald Chi-Square	Pr > ChiSq	
Intercept		1	0.2641	0.0190	0.2269	0.3014	192.83	<.0001	
B117_index		1	0.1178	0.0191	0.0803	0.1552	38.00	<.0001	
Age, Primary Case	Age_index_10	0-10	1	0.1688	0.0376	0.0951	0.2425	20.15	<.0001
	Age_index_10	20-30	1	-0.0032	0.0234	-0.0491	0.0428	0.02	0.8925
	Age_index_10	30-40	1	0.1488	0.0252	0.0994	0.1982	34.89	<.0001
	Age_index_10	40-50	1	0.1663	0.0250	0.1174	0.2152	44.40	<.0001
	Age_index_10	50-60	1	0.1780	0.0250	0.1289	0.2271	50.51	<.0001
	Age_index_10	60-70	1	0.2028	0.0303	0.1435	0.2622	44.87	<.0001
	Age_index_10	70-80	1	0.2111	0.0408	0.1312	0.2911	26.77	<.0001
	Age_index_10	80-90	1	0.2984	0.0582	0.1843	0.4125	26.27	<.0001
	Age_index_10	10-20	0	0.0000	0.0000	0.0000	0.0000	.	.
	Scale		0	1.0000	0.0000	1.0000	1.0000		

Log Likelihood	-3,440
AIC	6,900

Observations	10,834
Clusters	5,241

Table S8: Logit Model excluding Ct values

Analysis Of Maximum Likelihood Parameter Estimates								
Parameter		DF	Estimate	Standard Error	Wald 95% Confidence Limits		Wald Chi-Square	Pr > ChiSq
Intercept		1	-1.0250	0.0957	-1.2127	-0.8374	114.63	<.0001
B117_index		1	0.5061	0.0788	0.3517	0.6606	41.25	<.0001
Age_index_10	0-10	1	0.7482	0.1616	0.4314	1.0651	21.43	<.0001
Age_index_10	20-30	1	-0.0174	0.1176	-0.2480	0.2131	0.02	0.8823
Age_index_10	30-40	1	0.6675	0.1169	0.4383	0.8966	32.59	<.0001
Age_index_10	40-50	1	0.7406	0.1157	0.5138	0.9675	40.95	<.0001
Age_index_10	50-60	1	0.7879	0.1157	0.5612	1.0146	46.39	<.0001
Age_index_10	60-70	1	0.8888	0.1341	0.6260	1.1517	43.92	<.0001
Age_index_10	70-80	1	0.9222	0.1737	0.5817	1.2627	28.18	<.0001
Age_index_10	80-90	1	1.2745	0.2440	0.7964	1.7527	27.29	<.0001
Age_index_10	10-20	0	0.0000	0.0000	0.0000	0.0000	.	.
Scale		0	1.0000	0.0000	1.0000	1.0000		

Age, Primary Case

Log Likelihood	-3,439
AIC	6,898

Observations	10,834
Clusters	5,241

Table S8: Linear Model including Ct values

		DF	Estimate	Standard Error	Wald 95% Confidence Limits		Wald Chi-Square	Pr > ChiSq	
	Intercept	1	0.2539	0.0249	0.2051	0.3028	103.86	<.0001	
	B117_index	1	0.1094	0.0207	0.0688	0.1499	27.93	<.0001	
Age, Primary Case	Age_index_10	0-10	1	0.1893	0.0428	0.1054	0.2732	19.56	<.0001
	Age_index_10	20-30	1	0.0062	0.0246	-0.0420	0.0544	0.06	0.8005
	Age_index_10	30-40	1	0.1623	0.0267	0.1100	0.2147	36.93	<.0001
	Age_index_10	40-50	1	0.1950	0.0268	0.1425	0.2474	53.01	<.0001
	Age_index_10	50-60	1	0.1932	0.0265	0.1411	0.2452	52.97	<.0001
	Age_index_10	60-70	1	0.2081	0.0336	0.1423	0.2740	38.43	<.0001
	Age_index_10	70-80	1	0.2535	0.0544	0.1468	0.3602	21.69	<.0001
	Age_index_10	80-90	1	0.4442	0.1250	0.1992	0.6892	12.63	0.0004
	Age_index_10	10-20	0	0.0000	0.0000	0.0000	0.0000	.	.
	Ct Value	CT_value_2	<18	1	0.2483	0.1380	-0.0222	0.5188	3.24
CT_value_2		18-20	1	0.1918	0.0512	0.0915	0.2921	14.06	0.0002
CT_value_2		20-22	1	0.0452	0.0344	-0.0223	0.1126	1.72	0.1896
CT_value_2		22-24	1	0.0501	0.0276	-0.0039	0.1041	3.31	0.0689
CT_value_2		24-26	1	0.0073	0.0257	-0.0431	0.0576	0.08	0.7776
CT_value_2		26-28	1	-0.0152	0.0248	-0.0638	0.0333	0.38	0.5382
CT_value_2		30-32	1	-0.0245	0.0265	-0.0765	0.0275	0.85	0.3558
CT_value_2		32-34	1	-0.0317	0.0297	-0.0899	0.0265	1.14	0.2859
CT_value_2		34-36	1	-0.0228	0.0449	-0.1109	0.0653	0.26	0.6125
CT_value_2		36-38	1	-0.2502	0.0617	-0.3711	-0.1293	16.45	<.0001
CT_value_2		28-30	0	0.0000	0.0000	0.0000	0.0000	.	.
Scale			0	1.0000	0.0000	1.0000	1.0000		

Log Likelihood	-2,708
AIC	5,455

Observations	8,762
Clusters	4,172

Table S8: Logit Model including Ct values

		DF	Estimate	Standard Error	Wald 95% Confidence Limits		Wald Chi-Square	Pr > ChiSq	
	Intercept	1	-1.0881	0.1221	-1.3274	-0.8487	79.36	<.0001	
	B117_index	1	0.4729	0.0873	0.3019	0.6440	29.37	<.0001	
Age, Primary Case	Age_index_10	0-10	1	0.8547	0.1834	0.4953	1.2141	21.72	<.0001
	Age_index_10	20-30	1	0.0320	0.1259	-0.2148	0.2789	0.06	0.7994
	Age_index_10	30-40	1	0.7398	0.1256	0.4936	0.9860	34.68	<.0001
	Age_index_10	40-50	1	0.8628	0.1252	0.6175	1.1081	47.52	<.0001
	Age_index_10	50-60	1	0.8655	0.1238	0.6228	1.1081	48.88	<.0001
	Age_index_10	60-70	1	0.9292	0.1490	0.6372	1.2211	38.91	<.0001
	Age_index_10	70-80	1	1.1050	0.2294	0.6554	1.5545	23.21	<.0001
	Age_index_10	80-90	1	1.8850	0.6032	0.7027	3.0673	9.77	0.0018
	Age_index_10	10-20	0	0.0000	0.0000	0.0000	0.0000	.	.
	Ct Value	CT_value_2	<18	1	1.0256	0.6864	-0.3198	2.3709	2.23
CT_value_2		18-20	1	0.8185	0.2220	0.3835	1.2536	13.60	0.0002
CT_value_2		20-22	1	0.2128	0.1461	-0.0735	0.4990	2.12	0.1452
CT_value_2		22-24	1	0.2286	0.1179	-0.0026	0.4597	3.76	0.0526
CT_value_2		24-26	1	0.0305	0.1130	-0.1910	0.2520	0.07	0.7874
CT_value_2		26-28	1	-0.0468	0.1089	-0.2603	0.1666	0.18	0.6672
CT_value_2		30-32	1	-0.1188	0.1200	-0.3541	0.1165	0.98	0.3223
CT_value_2		32-34	1	-0.1402	0.1346	-0.4041	0.1236	1.09	0.2975
CT_value_2		34-36	1	-0.1163	0.2053	-0.5187	0.2861	0.32	0.5711
CT_value_2		36-38	1	-1.0109	0.3843	-1.7641	-0.2578	6.92	0.0085
CT_value_2		28-30	0	0.0000	0.0000	0.0000	0.0000	.	.
Scale			0	1.0000	0.0000	1.0000	1.0000		

	-
Log Likelihood	2,707
AIC	5,455

Observations	8,762
Clusters	4,172

R3: 6. In Figure 1, the transmission rate and transmission risk among different age groups might be confounded by other factors, e.g., age of their contacts. Please add age-specific transmission matrix, including: 1) number of primary cases and their potential secondary cases by age group, and 2) number of primary cases and their secondary cases by age group.

AU: We fully agree that the transmissibility of a primary case is dependent of the susceptibility of the potential secondary cases, e.g., age. Below, we have included the transmission rate for each lineage stratified by age of the primary case and age of the potential secondary cases, including the standard errors clustered at the household level. Note, that we have to group them to have precision in the estimates. Number of primary cases along with the number of potential secondary cases and positive secondary cases are included in table “Table S4 Summary statistics split by primary case age and their associated secondary cases”.

Figure: Age-by-age Transmission Rate by lineage

Notes: Standard errors clustered on the household level in parentheses.

R3: 7. The visualization in this paper needs to be improved.

R3: 7.1) Fig. 1 and Fig. S7-S9 show transmission rate and transmission risk by age, lineage and Ct value, but I'd also like to see absolute numbers of infected primary cases, potential secondary cases and secondary case by age, lineage and Ct value. I suggest the authors give these numbers for each key node in Fig. 1 and Fig. S7-S9.

AU: We think that adding this information is informative. Hence, we have added table S3 and S4.

R3: 7.2) Besides the observed transmission rate and transmission risk, one additional figure could be added to show the predicted probability of infection by age and lineage using generalized linear mixed model.

AU: We are not sure that we fully understand the reviewers suggestion. For estimating a random effects model, we have to assume exogeneity of the unobserved individual heterogeneity with respect to the individual (unobserved) heterogeneity. We have clustered standard errors on the household/primary case level, and handled this in that way. Therefore, we have not added any additional results here.

R3: 7.3) I suggest the authors enable lines on x-axis in Fig. S4-S6.

AU: We have added vertical grid lines on the x-axis in Figure S4, S6, S7 and S8, as requested. (Note, Figure S4-S6 in the original version is now Figure S4, S7, S8.)

R3: 8. Please specify the proportion of households with “co-primary cases” and please repeat this random process for several times to account for the uncertainty of transmission.

AU: We have specified the number and proportion of households with co-primary cases in line 179-181: “We furthermore performed our main analyses excluding households with co-primary cases (405 households, 5%) to investigate the sensitivity to misclassification of primary cases.” We agree that it is good to check the robustness of the results to the assignment of co-primary cases. Therefore, we have performed our main analyses entirely excluding households with co-primary cases. We have performed the analyses in Figure 1, Table 3, and Figure S10 excluding households with co-primary cases. The results are added as additional robustness analyses as Table S6, Figure S12, Figure S13.

Minor comments:

R3: 1. I suggest the authors add 95% confidence interval for secondary attack rates in Table 1, Table 2 and Table S2, and give coefficients with the p-value and odds ratio (OR) for each covariate in Table 3.

AU: Thank you for the comments. We have added 95%-CIs clustered on the household level for the attack rates in Table 1, 2, S2, S3, S4. The estimated fixed effects coefficients along with the p-values are included in the extended Table 3 below. However, we do not believe that this extension adds to the general readability of Table 3 and therefore suggest that we do not change our Table 3. If the editor wants to include the fixed effects estimates, we suggest including the extended Table 3 in the appendix.

Extended Table 3: Odds ratio estimates for transmissibility for B.1.1.7 compared with other lineages

	Transmission Rate				Transmission Risk		
	I	II	III	IV	V	VI	VII
B.1.1.7	1.50	1.68	1.69	1.63	1.52	1.66	1.61
95%-CI	(1.30-1.72)	(1.46-1.94)	(1.47-1.95)	(1.39-1.91)	(1.31-1.77)	(1.42-1.93)	(1.36-1.90)
Constant	1.00	1.00	1.00	1.00	1.00	1.00	1.00
	(<.0001)	(<.0001)	(<.0001)	(<.0001)	(<.0001)	(<.0001)	(0.0513)
Age, Primary Case							
0-10		1.62	1.65	1.68		2.10	2.32
		(0.0838)	(0.8578)	(0.5787)		(0.5460)	(0.8340)
10-20		0.84	0.81	0.75		0.98	1.03
		(<.0001)	(<.0001)	(<.0001)		(<.0001)	(<.0001)
20-30		1.00	1.00	1.00		1.00	1.00
		(.)	(.)	(.)		(.)	(.)
30-40		1.49	1.52	1.53		1.95	2.10
		(<.0001)	(0.2161)	(0.1127)		(0.9810)	(0.5175)
40-50		1.69	1.71	1.79		2.10	2.37
		(0.0260)	(0.8157)	(0.8339)		(0.3020)	(0.5792)
50-60		2.28	2.03	2.04		2.19	2.38
		(0.0230)	(0.0136)	(0.3201)		(0.0928)	(0.5574)
60-70		3.22	2.49	2.51		2.44	2.55
		(<.0001)	(0.0001)	(0.0181)		(0.0144)	(0.2950)
70-80		3.25	2.10	2.40		2.52	3.02
		(0.0001)	(0.1556)	(0.2082)		(0.0581)	(0.1312)
80-90		5.17	2.94	5.42		3.58	6.58
		(<.0001)	(0.0303)	(0.0704)		(0.0027)	(0.0390)
Age, Pot. Sec. Case							
0-10			0.96	0.97			
			(<.0001)	(<.0001)			
10-20			0.92	0.93			
			(<.0001)	(<.0001)			
20-30			1.00	1.00			
			(.)	(.)			
30-40			1.04	1.01			
			(<.0001)	(<.0001)			
40-50			1.10	1.18			
			(<.0001)	(<.0001)			
50-60			1.25	1.33			
			(<.0001)	(<.0001)			
60-70			1.46	1.41			

				(<.0001)	(<.0001)		
70-80				1.71	1.62		
				(<.0001)	(<.0001)		
80-90				2.09	1.90		
				(<.0001)	(0.0003)		
90-100				<0.001	<0.001		
				(<.0001)	(<.0001)		
Ct Value							
16-18					3.46		2.79
					(0.0195)		(0.1208)
18-20					2.08		2.21
					(0.0006)		(0.0003)
20-22					1.33		1.24
					(0.1988)		(0.3194)
22-24					1.30		1.26
					(0.1650)		(0.1639)
24-26					1.07		1.03
					(0.5855)		(0.6699)
26-28					0.97		0.95
					(0.0856)		(0.2203)
28-30					1.00		1.00
					(.)		(.)
30-32					0.87		0.89
					(0.0121)		(0.0809)
32-34					0.93		0.87
					(0.0899)		(0.0815)
34-36					0.92		0.89
					(0.2598)		(0.3076)
36-38					0.41		0.37
					(0.0039)		(0.0011)
Observations	10,834	10,834	10,834	8,762	10,834	10,834	8,762
Households	5,241	5,241	5,241	4,172	5,241	5,241	4,172

Notes: Columns I-IV provide odds ratio estimates for the increased transmission rate of B.1.1.7 compared with other lineages. Columns V-VII show the same for the transmission risk. Column I provides the crude estimates, i.e., only with a constant and without any controls. Column II further includes fixed effects for ten-year age groups of the primary cases. Column III further includes the age of potential secondary cases. Column IV further includes fixed effects for Ct values in bi-value groups. This sample is further restricted to only include primary cases identified in TCDK, as we only have Ct values on those. Column V provides the crude estimates, i.e., only with a constant and without any controls. Column VI further includes fixed effects for ten-year age groups of the primary cases. Column VII further includes fixed effects for Ct values in bi-value groups. This sample is further restricted to only include primary cases identified in TCDK, as we only have Ct values on those. All effects are included as fixed effects. Pot. Sec. Case = Potential Secondary Cases. Only primary cases identified in TCDK are included in models with Ct values. 95% confidence bands clustered on the household level. For the fixed effect estimates, p-values are in parenthesis.

R3: 2. The title of figures should be placed at the bottom of each figure

AU: We have moved all titles to the bottom of each figure.

R3: 3. “SARSCoV-2” on page 5 line 109 should be revised as “SARS-CoV-2”.

AU: Corrected.

R3: 4. “Supplementary Table S5” on page 8 line 178 should be revised as “Supplementary File”.

AU: Corrected.

REVIEWERS' COMMENTS

Reviewer #3 (Remarks to the Author):

The authors have provided clear explanations and included new analyses to clarify the ambiguity of the original manuscript, and most of my comments have been addressed properly. However, the large uncertainties remain regarding the estimated transmissibility of SARS-CoV-2 lineage B.1.1.7, due to a lack of detailed information on the generation of transmissions and exposure history. Thus, I suggest authors should at least add one paragraph to consider the potential impact of data availability on the estimated transmissibility in discussion section. Moreover, "Figure: Age-by-age Transmission Rate by lineage" shown in response letter should be included in appendix, which would give more information regarding age-specific susceptibility and infectivity to readers. In addition, numerator and denominator of transmission rate/risk in such figure should be given accordingly.

Reviewer #3 (R3)

Reviewer #3 (Remarks to the Author):

R3: The authors have provided clear explanations and included new analyses to clarify the ambiguity of the original manuscript, and most of my comments have been addressed properly. However, the large uncertainties remain regarding the estimated transmissibility of SARS-CoV-2 lineage B.1.1.7, due to a lack of detailed information on the generation of transmissions and exposure history. Thus, I suggest authors should at least add one paragraph to consider the potential impact of data availability on the estimated transmissibility in discussion section. Moreover, “Figure: Age-by-age Transmission Rate by lineage” shown in response letter should be included in appendix, which would give more information regarding age-specific susceptibility and infectivity to readers. In addition, numerator and denominator of transmission rate/risk in such figure should be given accordingly.

AU: Thank you for the comments.

We have included the following “Moreover, uncertainties regarding the heterogeneous transmissibility across lineage B.1.1.7 and other circulating lineages are present, as we did not have data on symptoms and exposure history.” in the discussion.

We have included the “Figure: Age-by-age Transmission Rate by lineage” as “Figure S14: Age-by-age Transmission Rate stratified by lineage” in the supplementary appendix and have included “Number of primary cases / number of potential secondary cases / number of positive secondary cases” below the standard errors.